# Dissemination and implementation of an evidence-based voluntary medical male circumcision program: The Spear & Shield program

Stephen M. Weiss[1], Kasonde Bowa[2], Robert Zulu[3,4], Violeta J. Rodriguez[1,5], Ryan R. Cook[6], Deborah L. Jones [1] *

1 Department of Psychiatry and Behavioral Sciences, University of Miami Miller School of Medicine, Miami, Florida, United States of America, 2 Clinical Sciences Department University of Lusaka, Lusaka, Zambia, 3 Ministry of Health, Provincial Health Office, Ndola, Copperbelt, Zambia, 4 Department of Health Promotion and Education, University of Zambia, School of Public Health, Lusaka, Zambia, 5 Department of Psychology, University of Georgia, Athens, Georgia, United States of America, 6 Medicine, General Internal Medicine, and Geriatrics, Oregon Health & Science University, Portland, Oregon, United States of America

* d.jones3@med.miami.edu

**Data Availability Statement:** Data is now available as a supplemental file.

## Abstract

Despite compelling evidence linking voluntary medical male circumcision (VMMC) with 60–70% HIV risk reduction in sub-Saharan Africa, Zambian men have been especially reluctant to undergo VMMC. The Government of Zambia set targets for VMMC uptake and promoted community-level interventions. Spear & Shield (S&S) is an innovative, evidence-based, service program promoting VMMC uptake while ensuring both VMMC supply and demand. This study assessed the large-scale provincial rollout of the program (S&S2) utilizing the RE-AIM model for translating interventions into the community. The S&S2 study was conducted between November 2015 and December 2020, and sequentially rolled out over four Zambian provinces in 96 clinics; 24 observation clinics received VMMC training only. Local clinic healthcare workers were trained to conduct the VMMC procedure and HIV counselors were trained to lead S&S group sessions. Using the RE-AIM model, primary outcomes were: Reach, the number, proportion, and representativeness of S&S attendees; Effectiveness, the impact of S&S2 on VMMC uptake; Adoption, the number, proportion, and representativeness of clinics implementing S&S2; Implementation, fidelity to the S&S intervention manual; and Maintenance, the extent to which S&S2 became an element of standard care within community clinics. Initially, $n = 109$ clinics were recruited; 96 were sustained and randomized for activation (Adoption). A total of 45,630 clinic patients ($n = 23,236$ men and $n = 22,394$ women) volunteered to attend the S&S sessions (Reach). The S&S2 program ran over 2,866 clinic-months (Implementation). Although the study did not target individual-level VMMCs, ~58,301 additional VMMCs were conducted at the clinic level (Effectiveness). Fidelity to the S&S intervention by group leaders ranged from 42%-95%. Sustainability of the program was operationalized as the number of CHCs initially activated that sustained the program. Intervention delivery ended, however, when study funding ceased (Maintenance). The S&S2 program successfully utilized the RE-AIM model to

**Funding:** This study was funded by National Institutes of Health/National Institute of Mental Health (NIH/NIMH) (R01MH095539), awarded to, DLJ, SMW, KB, RZ, and VJR with support from the University of Miami Miller School of Medicine Center for AIDS Research funded by NIH/National Institute of Allergy and Infectious Diseases (NIAID) (P30AI073961) awarded to DLJ. VJR's work on this study was also supported by a Ford Foundation Fellowship, administered by the National Academies of Sciences, Engineering, and Medicine (NASEM), a PEO Scholar Award from the PEO Sisterhood, and NIMH R36MH127838. The funding agencies were not involved in the writing of this manuscript or the decision to submit.

**Competing interests:** The authors have declared that no competing interests exist.

achieve study goals for implementation and dissemination in four Zambian provinces. Innovative VMMC programs such as S&S2 can improve the uptake of VMMC, one of the most effective strategies in the HIV prevention arsenal.

## Introduction

Voluntary medical male circumcision (VMMC) could prevent 3.4 million human immunodeficiency virus (HIV) infections in sub-Saharan Africa across a 10-year span and is an effective strategy for HIV prevention in high-prevalence populations [1–3]. Approximately 29.6 million VMMCs have been performed by 2020, averting 615, 000 new HIV infections. With the 29.6 million VMMCs conducted by 2020, if we stop circumcising now, 4.9 million new HIV infections will be averted by 2030 [4]. VMMC is also associated with reduced sexually transmitted infections (STIs), e.g., syphilis, human papilloma virus, cervical cancer, and urinary tract infections [1, 3, 5]. VMMC promotion began in Zambia in 2008; uptake was initially low due to lack of demand. In 2009, despite the high levels of protection from HIV associated with VMMC, ~80% of uncircumcised men in Zambia expressed no interest in undergoing VMMC [6, 7]. Models have shown that even with the scale-up of HIV treatment, VMMC remains a cost-effective, often cost-saving, prevention intervention in sub-Saharan Africa for at least the next 5 years [8].

The national promotion of VMMC in 2012 was an extended, intensive campaign designed to stimulate VMMC uptake, while ensuring VMMC supply matched demand [9]. In 2018, Zambians living with HIV numbered 1.2 million, HIV prevalence was 11.3% and 48,000 new infections were reported annually [10]. Despite high HIV prevalence, VMMC uptake in Zambia remained below 50% in many districts [11], lagging behind the President's Emergency Plan for Acquired Immune Deficiency Syndrome (AIDS) Relief's 80% saturation goal for priority countries [12]. Maintaining momentum during the Coronavirus Disease 2019 (COVID-19) pandemic in 2020; 227,824 men underwent VMMC and coverage increased by 15% [13]. In February 2021, 3,000,553 men in Zambia had been circumcised, representing 31% of the eligible population [11, 14]. Yet, these rates continued to fall short of the new 95% coverage goal established by the Zambian Operational Plan [15].

Behavioral change communication and demand generation, e.g., mass media and campaign materials that drive awareness and/or interest in VMMC through health promotion [11], are most effective in reaching those already contemplating VMMC [16]. Sensitization, the process through which those unaware of VMMC or its health benefits become aware [11] is the cornerstone of innovative demand generation. Suboptimal VMMC demand arises from several sources, e.g., fear of loss of sexual potency and concerns about partner preferences are key reasons to avoid VMMC [12, 17]. To stimulate VMMC demand and maximize its benefits, innovative community-level interventions are needed. The Spear & Shield (S&S) biobehavioral intervention was designed as a service program to increase VMMC uptake among Zambian men by training community health centers staff to stimulate demand through increased awareness of VMMC health benefits and safety among men and women [5, 18, 19]. The S&S randomized clinical trial targeted Zambian men from community health centers (CHCs) in the capital, Lusaka, who initially expressed no interest in undergoing VMMC. The S&S intervention consisted of four, weekly 90-minute group sessions on STIs, condom use, and VMMC; sessions utilized cognitive behavioral strategies to challenge thoughts leading to emotions that were barriers to undergoing VMMC. Men in S&S were twice as likely to undergo VMMC as the control group, and eight times more likely to undergo VMMC than the observation only condition. Men in S&S also reported greater condom use and increased sexual satisfaction [5].

Building on the S&S clinical trial, this study conducted a large-scale provincial rollout of S&S2 in Zambia, utilizing the RE-AIM implementation and dissemination framework for translating interventions into the community [20]. Specifically, the RE-AIM framework is comprised of the following components, 1) **R**each: the number, proportion, and representativeness of individuals willing to participate in the program, 2) **E**fficacy/Effectiveness: the impact of the intervention on important outcomes such as VMMC and potential negative effects, 3) **A**doption: the absolute number, proportion, and representativeness of settings or practices who are willing to incorporate the intervention into their health care delivery program, 4) **I**mplementation: a) at the practice level, the institution's fidelity to the core elements of a program, including consistency of delivery, and b) at the individual level, the patient's uptake of the intervention, e.g., S&S + VMMC, and 5) **M**aintenance: the extent to which a program becomes institutionalized as part of routine "standard of care." Using the RE-AIM model to achieve study goals for implementation and dissemination, S&S2 trained CHC staff to implement the S&S2 program in their CHCs [18, 19, 21] applying the Training of Trainers Model (previously described [22]). The S&S2 program was sequentially rolled out at the provincial level; our previous studies describe a series of mixed methods design studies guided by the Consolidated Framework for Implementation (CFIR) [18, 19, 21] specifically examining constructs associated with implementation outcomes (attendance). Given the RE-AIM framework, the objectives were to maximize the **R**each of S&S attendees; maximize the **E**ffectiveness of S&S by increasing the number of VMMCs; increase **A**doption, defined as the number, proportion, and representativeness of clinics implementing S&S2; achieve **I**mplementation, as defined by the fidelity to the S&S intervention manual; and **M**aintenance, the extent to which S&S2 became an element of standard care within community clinics. The current manuscript presents the final outcomes of the S&S2 program, describing the RE-AIM outcomes, including the circumcision rates overtime by condition in four Zambian provinces.

## Methods

### Ethical approval

All procedures performed in studies involving human participants were in accordance with the ethical standards of the institutional and/or national research committee and with the 2013 Declaration of Helsinki (Fortaleza, Brazil) and its later amendments or comparable ethical standards [23]. Prior to study initiation, investigators obtained approval from the Research Ethics Committee in Zambia (University of Zambia, approval number 002-11-15), the National Health Research Authority of Zambia (NHRA001/01/06/2023) from the provincial and district leadership, and from the affiliated US Institutional Review Board (University of Miami School of Medicine, approval number 20150710).

This study includes human participants; written informed consent was obtained from all participants prior to all study activities.

### Design

**Adoption.**   The S&S2 program was conducted in Lusaka, Central, Southern, and Copperbelt Provinces. Provinces were activated sequentially over four years, enabling the implementation stage of each province to be informed by earlier provinces. As provinces were activated, provincial and district health leaders were briefed on the study objectives and procedures. Eligible Community Health Centers (CHCs) had available space for VMMC, staff to train to provide VMMC and S&S sessions, and high HIV testing service (HTS) rates, and were selected with district guidance. Thirty CHCs per province were randomized into immediate (12) and delayed (12) activation or six observation only conditions. All CHCs received VMMC supplies

and VMMC training. Following province activation, in immediate activation CHCs, S&S intervention training was provided. After six months, the conclusion of the training period for immediate activation CHCs, S&S training was provided in delayed activation CHCs (see Fig 1). CHC inclusion criteria included: (1) conducting more than 50 HTS per month; (2)

**PROVINCE STAGGERED SEQUENTIAL IMPLEMENTATION FLOWCHART**

**Fig 1. Sequential rollout of provinces and training sequence with supervision for group leaders and VMMC training.** Caption. S&S = Spear & Shield.

located in a catchment area with greater than 10,000 residents; (3) sufficient space to conduct VMMC and S&S programs; and (4) available staff to conduct VMMC and S&S services and availability for training.

## Procedures

**Randomization.**  Prior to randomization, CHCs were stratified by catchment population to ensure comparable sized comparisons. The randomization of CHCs for activation was conducted by Zambian investigators not affiliated with the study using a random allocation computer-generated sequence; trial statisticians did not participate in randomization. CHCs were randomized using a staged randomization procedure in each province based on the 12 largest, 6 smallest, and 12 middle-size CHCs. In stage 1, the 12 largest CHCs and the 12 middle-size CHCs were randomized with random ordering of assignment (i.e., random ordering of the immediate, delayed, and observation only conditions). In stage 2, the smallest 12 CHCs were similarly allocated to condition.

**Training.**  S&S intervention (see also Bowa et al., 2022). Trainees were lay counselors > 18 years old engaged in HIV prevention services in the Government of Republic of Zambia. Clinics selected HTS counselors (one male, one female) to attend S&S training, plus two back-up trainees per site. Training consisted of a two-day intensive manualized workshop on the S&S intervention and study protocol, rigorous On the Job Training (OJT), and skill building to train future group leaders. Training included participatory strategies, e.g., role plays depicting real life situations that addressed myths, misconceptions and knowledge gaps on VMMC for HIV prevention [22]. Trainees then enrolled CHC attendees (i.e., three groups of 8–10 men and women) for OJT. Trainees co-led three groups of four sessions supervised by a S&S2 staff trainer: Group 1, S&S trainer conducted the session, leader observed; Group 2, session topics were divided between trainer and leader; Group 3, leader conducted sessions, trainer observed. Group leaders recruited for S&S from CHC HTS patients, the Out-Patient Department, youth friendly corners, and Maternal and Child Health services. A random sample of 10% of audio-recorded sessions were reviewed for intervention fidelity by the Zambia coordinator and a United States (US) investigator.

CHCs that had one or no VMMC providers selected two healthcare workers to train as VMMC providers per Zambian Ministry of Health VMMC guidelines. S&S2 VMMC provider trainees completed a 10-day, VMMC surgical training 3 months before the immediate S&S intervention training began. Training followed the WHO VMMC training manual and utilized the dorsal slit method as recommended by the Surgical Society of Zambia and Ministry of Health [24].

Conditions. Following Province activation, 12 CHCs randomized to the immediate intervention condition provided the S&S2 + VMMC program and 12 delayed intervention condition CHCs provided VMMC only. After 6 months, the delayed intervention condition clinics were trained to begin provision of the S&S + VMMC Program. Once implemented, CHCs continued to offer the S&S2 + VMMC program until completion of the study.

## Statistical analyses

Monthly CHC-level data, i.e., numbers of persons attending the clinic undergoing HTS and the total monthly VMMCs conducted, were collected at all sites to assess the influence of the ongoing intervention on overall clinic-level VMMC uptake. Monthly CHC-level VMMC rates were modeled using mixed negative binomial regression with month, study condition, and their interaction as the main predictors of interest, controlling for province and including the (log) number of HTS as an offset term. Highly variable and nonlinear time trends in VMMC

rates were modeled with cubic b-splines. CHCs were included as random intercepts in the model; although heterogeneity in CHC-specific time trends was likely, models including more complex random effects structures did not converge.

Hypotheses were tested by comparing model-predicted VMMC rates between VMMC + SS (i.e., after S&S training had been completed and sessions had been started), VMMC only (i.e., the 6-month period in CHCs randomized to the delayed condition where VMMC training and supplies were provided, but no S&S sessions or training), and observation-only conditions. Comparisons were made during two time periods, the entire study duration and the six months following study implementation. The average across the predicted monthly VMMC rates was computed within each time period, and rate ratios and confidence intervals were calculated to compare average rates across conditions. The first-six-month period was chosen to increase comparability between VMMC + SS and VMMC only conditions as the latter was active for a maximum of six months in each CHC. Several factors were likely to increase VMMC rates immediately following implementation, including newly trained, enthusiastic, and available intervention facilitators and medical professionals, adequately stocked supplies, and a cohort of men in the community eager to undergo VMMC for HIV prevention.

Measures of public health impact, including differences in absolute numbers of expected VMMCs per condition, were calculated by applying rate differences to the average number of HTS per clinic per month. Analyses were repeated in a "per-protocol" population, where clinic-months when no Spear & Shield sessions were conducted were reclassified as VMMC only. Finally, the additional impact of Spear & Shield women's groups was estimated by splitting VMMC + SS months into those where women's groups were active vs. those where only men's groups were being conducted. All analyses were completed using R v.4.05 with the 'geepack,' 'ggplot2,' 'lme4,' 'splines,' and 'emmeans' packages.

## Results

In summarizing the results, the results were presented using the RE-AIM framework. Specifically, the goals were to enhance the extent of Reach for S&S participants, improve the Effectiveness of S&S by increasing the number of VMMCs, boost Adoption by encouraging more clinics to implement S&S2 while considering the quantity, proportion, and representativeness of these clinics, ensure proper Implementation according to the S&S intervention manual and circumcision rates over time in four Zambian conditions, and promote Maintenance by integrating S&S2 as a standard practice in community clinics.

### Community health center and staff characteristics

Characteristics of CHC staff ($n$ = 385) assessed at all clinics are presented in Table 1. Table 2 presents monthly HTS and VMMC rates by study condition.

### Community health center recruitment and activation

**Adoption.**    Initially, $k$ = 109 CHCs were recruited into the study, and of these $k$ = 13 were deactivated (6 delayed and 7 immediate), leaving $k$ = 96 activated CHCs [19]. Of these 13, $k$ = 9 CHCs were deactivated due to failure to recruit male attendees into groups, $k$ = 1 due to an inability to hire an MC provider, $k$ = 1 due to limited resources, and $k$ = 2 due to lack of interest in adopting the program. Staff retention ranged from 25–100% (Central).

**Table 1. Clinic staff characteristics (N = 385).**

| | M (SD) (N %) |
|---|---|
| **Employed at facility (years)** | |
| 0–1 year | 75 (19.5%) |
| 2–3 years | 99 (25.7%) |
| 4–5 years | 69 (17.9%) |
| 6–7 years | 65 (16.9%) |
| 8–9 years | 25 (6.5%) |
| More than 10 years | 52 (13.5%) |
| **Job title** | |
| Physician/Clinic Officer/Sister in Charge | 56 (14.5%) |
| Professional Nurse | 103 (26.8%) |
| Assistant Nurse | 7 (1.8%) |
| Counsellor | 184 (47.8%) |
| Lay Health Worker | 35 (9.1%) |
| **Time in current position (years)** | |
| 0–1 year | 57 (14.8%) |
| 2–3 years | 79 (20.5%) |
| 4–5 years | 52 (13.5%) |
| 6–7 years | 67 (17.4%) |
| 8–9 years | 49 (12.7%) |
| More than 10 years | 81 (21.0%) |
| **Gender** | |
| Male | 181 (47.0%) |
| Female | 204 (53.0%) |
| **Age (years)** | 37.37 (9.6%) |
| **Education** | |
| Grade 8 –Grade 9 | 28 (7.3%) |
| Grade 10 | 10 (2.6%) |
| Grade 11 | 15 (3.9%) |
| Grade 12 | 83 (21.6%) |
| Diplomas/Occupational Certificates | 214 (55.6%) |
| First Degrees/Higher Diploma | 35 (9.1%) |
| **Monthly Income (Zambian Kwacha ~ USD)** | 3,044.38 (2072.65) ~176.99 USD |

**Table 2. Observed average monthly HTS and VMMC rates by study condition, over entire study period and within the first six months of intervention implementation.**

| | Entire study | | | First six months | | |
|---|---|---|---|---|---|---|
| | **HTS** | **VMMCs** | **Rate** | **HTS** | **VMMCs** | **Rate** |
| VMMC + SS | 853.5 | 101.9 | 0.2 | 535.4 | 102.3 | .26 |
| VMMC only | 447.3 | 76.5 | 0.22 | 447.3 | 76.5 | .22 |
| Observation | 553.1 | 63.5 | 0.19 | 311.6 | 44.0 | .18 |

*Note*. HTS = HIV testing services. VMMC = voluntary medical male circumcision. S&S = Spear & Shield.

### Enrolment and retention in S&S groups

**Reach.** S&S sessions were provided as a service program during regular clinic hours with no attendee compensation; attendees received a certificate of attendance. A total of 45,630 Zambians ($n$ = 23,236 men and $n$ = 22,394 women) attended the S&S groups. Male attendee retention across CHCs ranged from 77% to 100% (average 94%) attending all four sessions, whereas female attendee retention ranged from 79% to 100% (average 95%).

### Quality control and assurance in S&S groups

**Implementation.** Quality assurance evaluation of S&S groups were conducted to ensure intervention fidelity. Facilitator checklists and intervention recordings from sessions with groups 3 and 4 by the Project Manager were reviewed. A total of 256 recordings from 128 group leaders were reviewed and a score ranging from A to D (85%-100% = A, 70%-84% = B, 60%-69% = C and below 60% = D) was ascribed by study staff consensus; scores included calculating the rate of topics addressed of the intended content during sessions. Rates varied across Provinces; Lusaka Province CHCs covered 42% to 83% of intended content, Central Province CHCs covered 52% to 96% of the intended content. Southern Province rates ranged from 69% to 94%, Copperbelt rates ranged from 63% to 95%. Feedback was provided to group leaders on strengths and gaps identified or areas for improvement.

### VMMC rates

**Effectiveness.** To evaluate the influence of the ongoing intervention on overall CHC-level VMMC uptake, the observed and model-fitted monthly CHC-level VMMC rates across study conditions were calculated. Over the entire study period, adjusted average monthly VMMC rates were 15.5, 17.3, and 12.2 per 100 HTS in VMMC + SS, VMMC only, and observation-only conditions, respectively. VMMC rates were higher in VMMC + SS (IRR = 1.27, 95% CI 1.06 to 1.53, $p$ = .011) and VMMC only conditions (IRR = 1.42, 95% CI 1.06 to 1.91, $p$ = .019) compared to observation only. There was no difference in VMMC rates between VMMC + SS and VMMC only ($p$ = .4). Overall, rates were higher in the first six months of intervention implementation: 22.1, 17.3, and 13.5 per 100 HTS in VMMC + SS, VMMC only, and observation-only conditions. During this time period, VMMC + SS CHCs had higher VMMC rates than observation only (IRR = 1.63, 95% CI 1.28 to 2.08, $p$ < .001), but there was no difference between VMMC alone and observation only ($p$ = .09) or VMMC + SS and VMMC alone ($p$ = .12). Results were similar in the per-protocol sample.

### VMMCs attributable to the S&S intervention

CHCs performed an average of 616 HTS per month. Applying the model-predicted VMMC rates (see Fig 2), this volume of HTS implies an extra 20.3 VMMCs per month per CHC implementing VMMC training plus the S&S intervention compared to observation only. Over the 2,866 CHC-months when the S&S intervention was actively running, it resulted in an estimated 58,301 additional VMMCs performed, compared to what would be expected with no S&S intervention or VMMC training. Results over the entire study period were not compared with VMMC training alone as this condition was only active for the first six months following clinic activation. However, if VMMC rates in S&S CHCs could have been sustained at the early-implementation rate (i.e., that which occurred during the first six months), it is estimated that 53 additional VMMCs would be performed per clinic per month, or 151,938 over the study period compared to observation-only and 29 per clinic per month, or 84,802 compared to VMMC training alone.

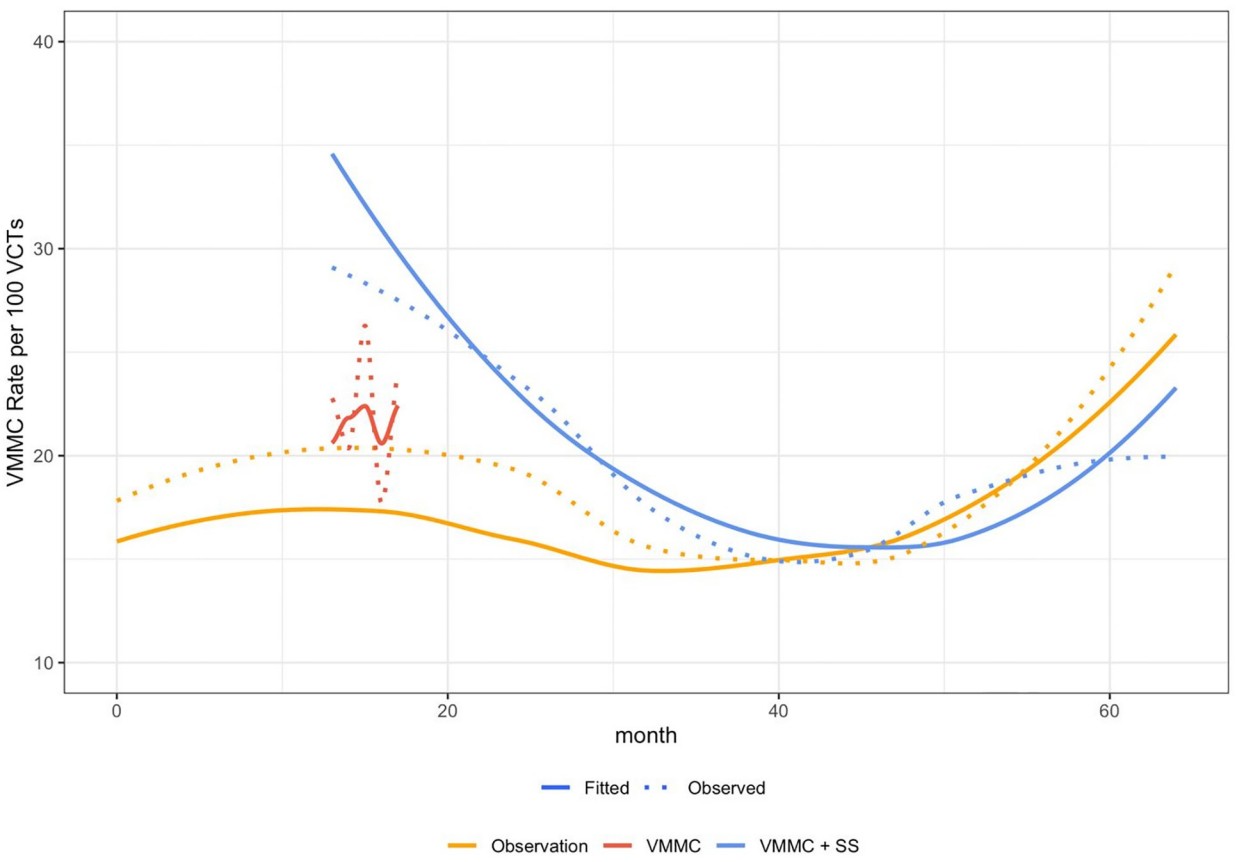

**Fig 2. VMMC rate by 100 HIV testing services performaned by condition.** Caption. S&S = Spear & Shield. VMMC = Voluntary Medical Male Circumcision. VCTs = Voluntary Counseling and Testing.

### Impact of women's groups

Average VMMC rates were highest during months when both S&S men's and women's groups were active: 15.6 per 100 HTS as compared to 12.7 with only men's groups, 14.8 with VMMC only, and 12.2 under no intervention (observation only). The rate ratio comparing S&S + women to no intervention was modest, 1.3 (95% CI 1.05 to 1.56, $p$ = .014), but there was no difference between S&S + women and other study conditions (Tables 3 and 4).

**Maintenance.** Sustainability of the program was operationalized as the number of CHCs initially activated that sustained the program. Intervention delivery ended, however, when study funding ceased.

## Discussion

This study presents the implementation and dissemination of the Spear & Shield 2 program using the RE-AIM model, building on previous research by this team [18, 19, 21, 22]. The S&S2 results indicate successful dissemination and implementation of the program across four provinces in Zambia, resulting in enhanced uptake of VMMC. Though the RE-AIM model has been used extensively in many settings, its use in Zambia has been limited, and results demonstrate the utility of the model in the Zambian context. This study thus demonstrates the successful implementation and dissemination of the S&S2 program in Zambia,

**Table 3. Fitted average monthly VMMC rates per 100 HTS, by study condition, over entire study period and within the first six months of intervention implementation, intent-to-treat and per-protocol samples.**

|  | Entire study | | | | First six months | | | |
|---|---|---|---|---|---|---|---|---|
|  | Rate | IRR | 95% CI | p | Rate | IRR | 95% CI | p |
| Intent to treat |  |  |  |  |  |  |  |  |
| VMMC + SS | 15.5 | 1.27 | (1.06, 1.53) | .011 | 22.1 | 1.63 | (1.28, 2.08) | < .001 |
| VMMC only | 17.3 | 1.42 | (1.06, 1.91) | .019 | 17.3 | 1.28 | (0.96, 1.71) | .091 |
| Observation | 12.2 | Ref |  |  | 13.5 | Ref |  |  |
| Per-protocol |  |  |  |  |  |  |  |  |
| VMMC + SS | 15.3 | 1.26 | (1.04, 1.52) | .021 | 21.6 | 1.6 | (1.18, 2.18) | .003 |
| VMMC only | 14.8 | 1.21 | (0.99, 1.49) | .063 | 19.6 | 1.45 | (1.16, 1.82) | .001 |
| Observation | 12.2 | Ref |  |  | 13.5 | Ref |  |  |

*Note.* HTS = HIV testing services. VMMC = voluntary medical male circumcision. S&S = Spear & Shield.

**Table 4. Fitted average monthly VMMC rates per 100 HTS.** VMMC + SS months are split into those where women's groups were conducted vs. those where only men's groups were running.

|  | Entire study | | | | First six months | | | |
|---|---|---|---|---|---|---|---|---|
|  | Rate | IRR | 95% CI | p | Rate | IRR | 95% CI | p |
| Intent to treat |  |  |  |  |  |  |  |  |
| VMMC + SS + Women | 15.6 | 1.28 | (1.05, 1.56) | .014 | 20.7 | 1.53 | (1.11, 2.10) | .008 |
| VMMC + SS | 12.7 | 1.05 | (0.78, 1.40) | .76 | 24.8 | 1.83 | (0.76, 4.78) | .216 |
| VMMC only | 14.8 | 1.21 | (0.98, 1.49) | .07 | 19.7 | 1.45 | (1.16, 1.82) | .001 |
| Observation | 12.2 | Ref |  |  | 13.5 | Ref |  |  |

Note. *Note.* HTS = HIV testing services. VMMC = voluntary medical male circumcision. S&S = Spear & Shield.

using the RE-AIM model as a framework. The results indicate that S&S2 effectively reached a significant number of individuals at risk for HIV infection and resulted in increased uptake of VMMC among men. The use of the RE-AIM model in the Zambian context proved to be valuable and showcased the utility of the model in evaluating program outcomes.

S&S2 successfully reached a substantial number of men and women at risk for HIV infection and enhancing VMMC uptake among men. Within the RE-AIM model, Reach represents the number, proportion, and representativeness of the S&S2 program attendees. Attendees were CHC patients drawn from HTS and as such were representative of those in Zambia who perceived themselves as at risk of HIV infection. The program successfully engaged (Reached) over 45,000 men and women, who were representative of those in Zambia perceiving themselves at risk of HIV infection. Effectiveness of the S&S2 program on VMMC uptake at the CHC level was associated with an estimated 58,301 additional VMMCs during S&S2, compared to CHCs without S&S2, supporting previous S&S results [5]. Finally, the goal of S&S2 was to translate this efficacious intervention into a service program utilizing local, trainable talent. VMMC CHC level uptake was impacted by VMMC campaigns in each province, and the secular trend and the observation only sites within each province indicated that VMMC campaigns activated in the last 18 months of the study moderated and augmented the S&S effect. The findings also highlighted the importance of VMMC campaigns and the support of clinic leadership in enhancing VMMC rates.

Adoption in the RE-AIM model refers to the number, proportion, and representativeness of CHCs implementing the S&S2 program (VMMC+SS), whereas Implementation refers to fidelity to the S&S intervention [20]. Adoption was illustrated by the high rate of adoption (88%) among activated CHCs throughout the sequential rollout of S&S2. Adoption was also evidenced by staff engagement, such that study retention rates among staff and attendees were relatively high in three of four provinces. Implementation, evaluated as fidelity to the intervention, was partially supported, with intervention fidelity rates ranging from 42% to 96% across the provinces. Factors associated with intervention delivery identified included having less dedicated time and space to conduct the intervention due to conflicting demands within the CHC, some of which arose from other public health campaigns. Results from our previous research also suggest that the most effective delivery of the intervention relied on support from clinic leadership [18, 19, 21]. In addition, VMMC rates were exceptionally high in the 6 months to one year following activation. Both fidelity data and VMMC uptake suggest that after one year, additional "booster sessions" may be needed to improve motivation among CHC staff. Incentives for undergoing VMMC [25], women's involvement [26] and community outreach promoting S&S after more interested candidates participate may also enhance demand for VMMC. As such, this study suggests the need for additional support, including booster sessions and incentives, to sustain motivation among clinic staff and increase demand for VMMC.

Maintenance is the extent to which S&S2 became an element of the "standard of care" within CHCs [20]. Only 12% of CHCs were unable to enroll clients or carry out group sessions and were subsequently deactivated due to failure to recruit, initiate and sustain group sessions. Even in Lusaka, the longest running province, only 5 CHCs had been deactivated by the end of the study. As such, the Maintenance of S&S2 was supported within the Zambian context. However, though CHC staff were enthusiastic and capable of achieving the goals of the intervention, the lack of modest funding to sustain the clinic personnel providing the program led to discontinuation of the S&S2 intervention at study end. Our study therefore highlights the need for ongoing efforts to institutionalize S&S2 as a standard of care in order to ensure its long-term sustainability.

This study's results must be interpreted within the context of clinic rather than individual-level data; results reflect the VMMC outcomes from the overall clinic, not of the men attending S&S sessions. As previously found [26], VMMC rates were higher when S&S2 men's and women's groups were active, although not statistically significant. The original trial found women's participation accounted for ~6% of VMMCs performed [26], which at the population level could account for a significant number of VMMCs [5]. The lack of differences in per protocol analyses may be attributable to high attendance and retention rates in men's and women's groups, and by the potential for women attending S&S groups to be partners of men not attending S&S. Though not evaluated, S&S may directly, rather than indirectly, benefit women participating, which may also contribute to high participation rates. Future studies may examine the impact of S&S on women and their experiences in attending S&S groups with or without a male partner.

## Limitations

Concurrent with successful implementation of S&S2 in Zambia, study limitations should be considered. VMMC data were limited to CHC level monthly counts rather than individual-level data. Collecting individual-level data would have increased CHC staff burden and constricted the attendance of the 45,000 men and women. However, as the design focused on the dissemination and implementation at the CHC level, CHC staff were evaluated at

the individual level, reducing CHC burden. This was an important design consideration for the already overburdened healthcare system in Zambia. However, this design did not account for potential influence of other demand creation activities happening in the area during the study period. VMMC + SS and VMMC only conditions were compared to observation-only clinics in order to account for secular trends in VMMC rates over the study period (Fig 2). However, the observation-only clinics were only a small subsample of all possible clinics in the four provinces, and other trends may not have been captured. For example, during the study period, one province initiated a campaign on a different health issue, seconding clinic staff for much of the study duration thereby reducing SS2 impact. Finally, study completion in December 2020 overlapped during the onset of the COVID-19 pandemic in Africa.

## Conclusion

Overall, this study highlights the importance of translating evidence-based interventions into practice and evaluating their real-world impact. Translation of an evidence-based intervention to practice, sustaining it, and evaluating the real-world evidence produced is a lengthy, complicated process. To translate the S&S intervention, study staff trained group leaders and VMMC providers in 96 clinics. During dissemination, challenges to implementation and sustainment were addressed and study staff provided ongoing support in the field [18, 19, 21]. Nearly 3600 months of VMMC data from over 100 CHCs plus 100 years of historical VMMC data from VMMCs performed during the year prior to study commencement was collected. Although individual-level VMMC rates were not targeted, the estimated 58,301 additional VMMCs conducted at the clinic level could have reached an estimated 151,938 if VMMC rates could have been sustained at the early-implementation level. This, in and of itself, is a major contribution to the national goals in HIV prevention. The sustainability of VMMC as a public health intervention now relies on its institutionalization within the health system as a component of HIV prevention services [11]. Continued expansion of VMMC programs, such as adolescent or infant circumcision, has the potential to further reduce the burden of future illness through prevention. Effective public health service programs remain an investment in the future . . . today.

## Supporting information

**S1 Data. Clinic data.**
(XLS)

**S1 Checklist. Inclusivity in global research.**
(DOCX)

## Author Contributions

**Conceptualization:** Stephen M. Weiss, Kasonde Bowa, Robert Zulu, Ryan R. Cook, Deborah L. Jones.

**Data curation:** Violeta J. Rodriguez, Deborah L. Jones.

**Formal analysis:** Violeta J. Rodriguez, Ryan R. Cook.

**Funding acquisition:** Stephen M. Weiss, Kasonde Bowa, Deborah L. Jones.

**Investigation:** Stephen M. Weiss, Kasonde Bowa, Robert Zulu, Violeta J. Rodriguez, Deborah L. Jones.

**Methodology:** Stephen M. Weiss, Kasonde Bowa, Robert Zulu, Violeta J. Rodriguez, Ryan R. Cook.

**Project administration:** Stephen M. Weiss, Robert Zulu.

**Resources:** Stephen M. Weiss.

**Supervision:** Stephen M. Weiss, Kasonde Bowa.

**Validation:** Stephen M. Weiss.

**Writing – original draft:** Stephen M. Weiss, Violeta J. Rodriguez, Ryan R. Cook, Deborah L. Jones.

**Writing – review & editing:** Stephen M. Weiss, Kasonde Bowa, Robert Zulu, Violeta J. Rodriguez, Deborah L. Jones.

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
