## [Decision Letter · Decision Letter 0]

19 May 2023

PGPH-D-23-00137

Dissemination and implementation of an evidence-based voluntary medical male circumcision program: The Spear & Shield Program

Dear Dr. Jones,

Thank you for submitting your manuscript to PLOS Global Public Health. After careful consideration, we feel that it has merit but does not fully meet PLOS Global Public Health’s publication criteria as it currently stands. Therefore, we invite you to submit a revised version of the manuscript that addresses the points raised during the review process.

We look forward to receiving your revised manuscript.

Kind regards,

Jonathan Huang

Academic Editor

Journal Requirements:

1. Please include information in the Methods section on type of consent obtained, and whether the IRBs approved the consent procedures." 2) “Please include a complete copy of PLOS’ questionnaire on inclusivity in global research in your revised manuscript. Our policy for research in this area aims to improve transparency in the reporting of research performed outside of researchers’ own country or community. The policy applies to researchers who have travelled to a different country to conduct research, research with Indigenous populations or their lands, and research on cultural artefacts. The questionnaire can also be requested at the journal’s discretion for any other submissions, even if these conditions are not met.  Please find more information on the policy and a link to download a blank copy of the questionnaire here: https://journals.plos.org/plosone/s/best-practices-in-research-reporting. Please upload a completed version of your questionnaire as Supporting Information when you resubmit your manuscript.

2. Please provide separate figure files in .tif or .eps format.

Additional Editor Comments (if provided):

Reviewers' comments:

Reviewer's Responses to Questions

**Comments to the Author**

1. Does this manuscript meet PLOS Global Public Health’s publication criteria? Is the manuscript technically sound, and do the data support the conclusions? The manuscript must describe methodologically and ethically rigorous research with conclusions that are appropriately drawn based on the data presented.

Reviewer #1: Partly

Reviewer #2: Yes

2. Has the statistical analysis been performed appropriately and rigorously?

Reviewer #1: No

Reviewer #2: I don't know

3. Have the authors made all data underlying the findings in their manuscript fully available (please refer to the Data Availability Statement at the start of the manuscript PDF file)?

Reviewer #1: Yes

Reviewer #2: No

4. Is the manuscript presented in an intelligible fashion and written in standard English?

Reviewer #1: Yes

Reviewer #2: Yes

5. Review Comments to the Author

Reviewer #1: This is a very interesting manuscript describing the scale up of VMMC in 4 provinces of Zambia using the S&S and RE-AIM framework. In each province, clinic were divided into immediate (12), delayed (12) and observation groups (6) regarding VMMC (observation) and S&S (immediate and delayed). Result shows an increase of VMMC in the immediate clinics.

General: Clarity and organization of the paper can be improved to easy understanding. Write in full first time an acronym is used. review typos and totals.

Specific:

Methods

Ethics

1. Please considered a reference to the last version of the "Declaration of Helsinki 2013, Fortaleza, Brazil"

2. Please specify "written or oral consent"?

Procedures

1."conditions" and "Training" on 2nd and 3rd paragraphs respectively seem dislocated.

2. "Trainees were GRZ lay counselors" please review sentence.

Results

1. Says "N= 310", Actual sum =385

2. Table 1 : Please reorganize: "Employment at facilite (years)" - Less than 1 year, [2-3] etc,

3. Table 1: Be clear on what is presented: Age in years (mean, Median, SD, IQR)

4. Table 1: Be clear on what is presented: Monthly income in USD/ZK (mean, Median, SD, IQR)

5. Tabel 1: Include all acronyms in full in the figure legend caption

6. Ranges provided (average) > This is not clear - Mean or medium?

7. Figures need legends captions

Discussion

1. Focus on results of this study not only the potential results.

Reviewer #2: Thank you for this opportunity to review this interesting article titled" Dissemination and implementation of an evidence-based voluntary medical male circumcision program: The Spear & Shield Program". The manuscript is technically sound and reaches a conclusion that is consistent with previous research. The study adds significant value to the global efforts aiming to contain HIV infections. Nonetheless, data supporting findings are required to be attached or deposited to a public repository.

6. PLOS authors have the option to publish the peer review history of their article (what does this mean?). If published, this will include your full peer review and any attached files.

**Do you want your identity to be public for this peer review?** For information about this choice, including consent withdrawal, please see our Privacy Policy.

Reviewer #1: No

Reviewer #2: No

---

## [Decision Letter · Decision Letter 1]

14 Jul 2023

PGPH-D-23-00137R1

Dissemination and implementation of an evidence-based voluntary medical male circumcision program: The Spear & Shield Program

Dear Dr. Jones,

Thank you for submitting your manuscript to PLOS Global Public Health. After careful consideration, we feel that it has merit but does not fully meet PLOS Global Public Health’s publication criteria as it currently stands. Therefore, we invite you to submit a revised version of the manuscript that addresses the points raised during the review process.

We look forward to receiving your revised manuscript.

Kind regards,

Jianhong Zhou

Staff Editor

Journal Requirements:

b. If any authors received a salary from any of your funders, please state which authors and which funders.

Additional Editor Comments (if provided):

Reviewers' comments:

Reviewer's Responses to Questions

**Comments to the Author**

1. If the authors have adequately addressed your comments raised in a previous round of review and you feel that this manuscript is now acceptable for publication, you may indicate that here to bypass the “Comments to the Author” section, enter your conflict of interest statement in the “Confidential to Editor” section, and submit your "Accept" recommendation.

Reviewer #1: All comments have been addressed

Reviewer #3: (No Response)

Reviewer #4: (No Response)

2. Does this manuscript meet PLOS Global Public Health’s publication criteria? Is the manuscript technically sound, and do the data support the conclusions? The manuscript must describe methodologically and ethically rigorous research with conclusions that are appropriately drawn based on the data presented.

Reviewer #1: Yes

Reviewer #3: (No Response)

Reviewer #4: No

3. Has the statistical analysis been performed appropriately and rigorously?

Reviewer #1: Yes

Reviewer #3: (No Response)

Reviewer #4: I don't know

4. Have the authors made all data underlying the findings in their manuscript fully available (please refer to the Data Availability Statement at the start of the manuscript PDF file)?

Reviewer #1: Yes

Reviewer #3: (No Response)

Reviewer #4: Yes

5. Is the manuscript presented in an intelligible fashion and written in standard English?

Reviewer #1: Yes

Reviewer #3: (No Response)

Reviewer #4: Yes

6. Review Comments to the Author

Reviewer #1: (No Response)

Reviewer #3: 1. Introduction needs updating.

Close to 30M VMMC had been performed by 2020, averting 615, 000 new HIV infections. With the 29.6M VMMC conducted by 2020, if we stop circumcising now, 4.9 M new HIV infections will be averted by 2030 (Uneven progress on the voluntary medical male circumcision. Programme across 15 eastern and southern African countries in the face of the COVID-19 pandemic. Geneva: UNAIDS & WHO; 2022).

Recently, at AIDS Impact, # of VMMC was reported to be >30M and even more new infections averted? (contact zembel@unaids.org for latest figures).

Models have shown that VMMC remains a cost-effective, often cost-saving, prevention intervention in sub-Saharan Africa for at least the next 5 years (Cost-effectiveness of voluntary medical male circumcision for HIV prevention across sub-Saharan Africa: results from five independent models. Lancet Glob Health. 2023). Concerns were whether it was still effective given scale-up of HIV treatment (for prevention and treatment) programs).

2. Ethical approval - include approvals numbers (e.g. Prior to study initiation, investigators obtained approval from the Research Ethics Committee in Zambia (University of Zambia, #XXX), from the provincial and district leadership, and from the affiliated US Institutional Review Board (University of Miami, #YYY).

3. Minor edits

Innovative VMMC programs such as S&S2 can improve the uptake VMMC, of one of the most...?

VMMC program until completion of the study (no full stop).

Reviewer #4: Summary

The manuscript describes the programmatic roll out of interventions derived from the Spear and Shield (S & S) clinical trial to four provinces in Zambia. S&S intervention was associated with increased likelihood to of circumcision uptake and higher rate of condom use. Its programmatic roll out was implemented from Nov 2015 to December 2020 to promote uptake of VMMC. Authors assessed the roll out using RE-AIM model to measure VMMC reach, uptake, and other attributes of program performance. They reported an increase in number of circumcisions at clinic level and concluded that the program successfully utilized the RE-AIM model to achieve study goals for implementation and dissemination.

General comments:

The manuscript raises important considerations for improving VMMC program update by matching demand creation with service availability. It however requires edits to improve clarity of the link Objectives, methods, results and conclusions. It would be helpful to start by crystalizing the objectives.

Specific Comments

Abstract:

The results do not clearly show how the RE-AIM model helped achieve the study goals for implementation and dissemination as stated in the conclusion. The authors should edit the abstract to show a clear link between objectives, RE-AIM Model, results and conclusion. The results of assessing the VMMC program roll out against each element of RE-AIM framework should be made clearer.

Statistical analysis

A found it difficult to align the different fragmentations of the analysis with the objectives and the RE AIM framework. I am not sure if it can be simplified for the readers.

7. PLOS authors have the option to publish the peer review history of their article (what does this mean?). If published, this will include your full peer review and any attached files.

**Do you want your identity to be public for this peer review?** For information about this choice, including consent withdrawal, please see our Privacy Policy.

Reviewer #1: No

Reviewer #3: No

Reviewer #4: No

---

## [Decision Letter · Decision Letter 2]

7 Nov 2023

PGPH-D-23-00137R2

Dissemination and implementation of an evidence-based voluntary medical male circumcision program: The Spear & Shield Program

Dear Dr. Jones,

Thank you for submitting your manuscript to PLOS Global Public Health. After careful consideration, we feel that it has merit but does not fully meet PLOS Global Public Health’s publication criteria as it currently stands. Therefore, we invite you to submit a revised version of the manuscript that addresses the points raised during the review process.

All of the reviewers are satisfied that your revisions have dealt with the concerns they raised earlier in the review process.

However, there is one request in the original decision letter that is still outstanding:

Please include a complete copy of PLOS’ questionnaire on inclusivity in global research in your revised manuscript. Our policy for research in this area aims to improve transparency in the reporting of research performed outside of researchers’ own country or community. The policy applies to researchers who have travelled to a different country to conduct research, research with Indigenous populations or their lands, and research on cultural artefacts. The questionnaire can also be requested at the journal’s discretion for any other submissions, even if these conditions are not met.  Please find more information on the policy and a link to download a blank copy of the questionnaire here: https://journals.plos.org/plosone/s/best-practices-in-research-reporting. Please upload a completed version of your questionnaire as Supporting Information when you resubmit your manuscript.

We look forward to receiving your revised manuscript.

Kind regards,

Steve Zimmerman, PhD

PLOS Staff Editor

Journal Requirements:

2. Please include information in the Methods section on type of consent obtained, and whether the IRBs approved the consent procedures." 2) “Please include a complete copy of PLOS’ questionnaire on inclusivity in global research in your revised manuscript. Our policy for research in this area aims to improve transparency in the reporting of research performed outside of researchers’ own country or community. The policy applies to researchers who have travelled to a different country to conduct research, research with Indigenous populations or their lands, and research on cultural artefacts. The questionnaire can also be requested at the journal’s discretion for any other submissions, even if these conditions are not met.  Please find more information on the policy and a link to download a blank copy of the questionnaire here: https://journals.plos.org/plosone/s/best-practices-in-research-reporting. Please upload a completed version of your questionnaire as Supporting Information when you resubmit your manuscript.

Additional Editor Comments (if provided):

Reviewers' comments:

Reviewer's Responses to Questions

**Comments to the Author**

1. If the authors have adequately addressed your comments raised in a previous round of review and you feel that this manuscript is now acceptable for publication, you may indicate that here to bypass the “Comments to the Author” section, enter your conflict of interest statement in the “Confidential to Editor” section, and submit your "Accept" recommendation.

Reviewer #3: All comments have been addressed

Reviewer #4: All comments have been addressed

2. Does this manuscript meet PLOS Global Public Health’s publication criteria? Is the manuscript technically sound, and do the data support the conclusions? The manuscript must describe methodologically and ethically rigorous research with conclusions that are appropriately drawn based on the data presented.

Reviewer #3: Yes

Reviewer #4: Yes

3. Has the statistical analysis been performed appropriately and rigorously?

Reviewer #3: Yes

Reviewer #4: Yes

4. Have the authors made all data underlying the findings in their manuscript fully available (please refer to the Data Availability Statement at the start of the manuscript PDF file)?

Reviewer #3: Yes

Reviewer #4: Yes

5. Is the manuscript presented in an intelligible fashion and written in standard English?

Reviewer #3: Yes

Reviewer #4: Yes

6. Review Comments to the Author

Reviewer #3: All comments have been addressed.

Reviewer #4: No further comments

7. PLOS authors have the option to publish the peer review history of their article (what does this mean?). If published, this will include your full peer review and any attached files.

**Do you want your identity to be public for this peer review?** For information about this choice, including consent withdrawal, please see our Privacy Policy.

Reviewer #3: No

Reviewer #4: No

---

## [Editor Report · Decision Letter 3]

20 Dec 2023

Dissemination and implementation of an evidence-based voluntary medical male circumcision program: The Spear & Shield Program

PGPH-D-23-00137R3

Dear Dr. Jones,

We are pleased to inform you that your manuscript 'Dissemination and implementation of an evidence-based voluntary medical male circumcision program: The Spear & Shield Program' has been provisionally accepted for publication in PLOS Global Public Health.

Best regards,

Julia Robinson

Staff Editor